# Learning a reactive potential for silica-water through uncertainty attribution

Swagata Roy[1], Johannes P. Dürholt[2], Thomas S. Asche [2], Federico Zipoli[3] & Rafael Gómez-Bombarelli [1] ✉

The reactivity of silicates in aqueous solution is relevant to various chemistries ranging from silicate minerals in geology, to the C-S-H phase in cement, nanoporous zeolite catalysts, or highly porous precipitated silica. While simulations of chemical reactions can provide insight at the molecular level, balancing accuracy and scale in reactive simulations in the condensed phase is a challenge. Here, we demonstrate how a machine-learning reactive interatomic potential trained on PaiNN architecture can accurately capture silicate-water reactivity. The model was trained on a dataset comprising 400,000 energies and forces of molecular clusters at the $\omega$B97X-D3/def2-TZVP level. To ensure the robustness of the model, we introduce a general active learning strategy based on the attribution of the model uncertainty, that automatically isolates uncertain regions of bulk simulations to be calculated as small-sized clusters. The potential reproduces static and dynamic properties of liquid water and solid crystalline silicates, despite having been trained exclusively on cluster data. Furthermore, we utilize enhanced sampling simulations to recover the self-ionization reactivity of water accurately, and the acidity of silicate oligomers, and lastly study the silicate dimerization reaction in a water solution at neutral conditions and find that the reaction occurs through a flanking mechanism.

One of the most abundant materials on the planet, silica[1] and its many polymorphs are used in applications like catalysts[2], pharmaceuticals[3], nanotechnology[4], and additives[5]. Amorphous silica, in particular, has dozens of commercial applications and the global precipitated silica market is expected to reach US\$3.49 billion by 2023[6].

Precipitated silica or silica gel is produced by the polymerization of water-soluble silicate salts in the presence of mineral acids, resulting in a precipitate or gel[7]. Similarly, zeolites are crystallized from a precursor gel in hydrothermal conditions[8]. While various experimental techniques can characterize nanosized crystallites and precipitates, they cannot resolve the individual silicate aggregates or the network of silicon connectivity[9,10].

Molecular simulations of reactivity in the aqueous condensed phase can bridge this knowledge gap on silica condensation. The key

challenge is describing covalent bond breaking and formation accurately while being able to simulate large length- and time-scales. Numerous classical interatomic potentials for anhydrous silica have been generated to simulate silica glasses or other polymorphs[11,12], including the well-known BKS[11] pair potential for $\alpha$-quartz and silica glasses, and more recent reparametrizations for nanoclusters and amorphous silica[12,13]. These models, along with ReaxFF models parameterized for water-silica[14,15] cannot accurately replicate the condensation reactions in aqueous solution. The recent advances in machine learning (ML) techniques and the expansion of neural networks (NN) and automatic differentiation routines[16,17] have provided a branch between physical sciences and statistical learning. NN-based interatomic potentials (NNIP) built on a variety of molecular representations retain the accuracy of the ab initio training data and can be

[1]Department of Materials Science and Engineering, Massachusetts Institute of Technology, Cambridge, MA, USA. [2]Evonik Operations GmbH, Essen, North Rhine-Westphalia, Germany. [3]IBM Research Europe, Saümerstrasse 4, 8803 Rüschlikon, Switzerland. ✉e-mail: rafagb@mit.edu

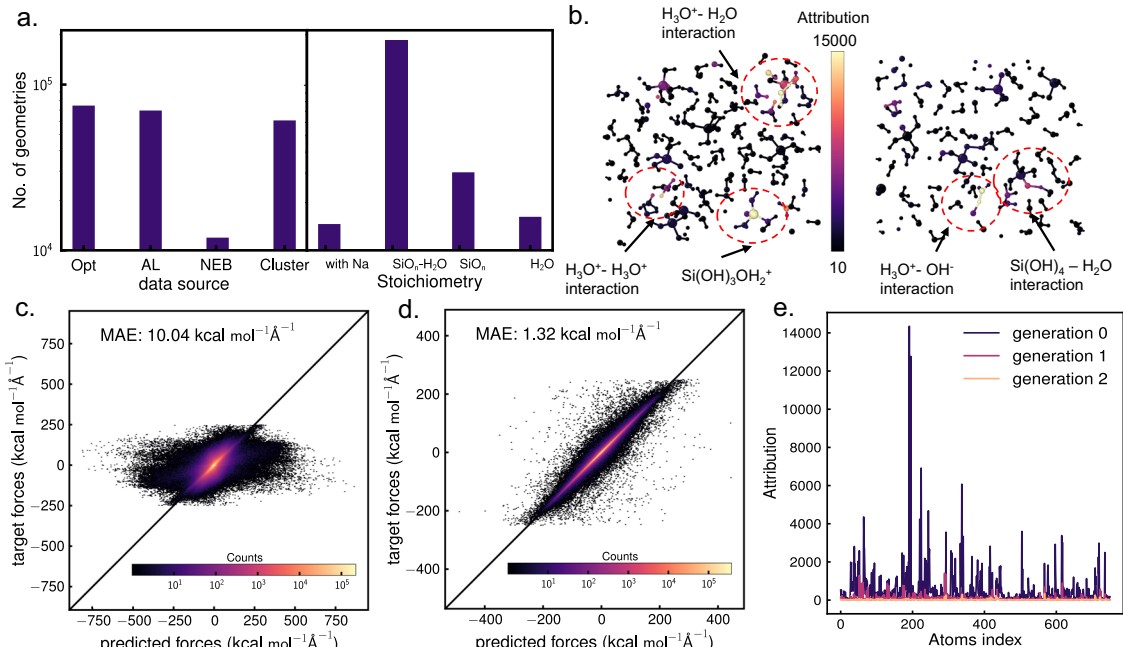

**Fig. 1 | Training dataset and active learning method. a** Distribution of data based on data source - Cluster: silicate and water clusters based on random position guesses. Opt: molecule clusters optimized at a lower level of Density Functional Theory (DFT) or Semiempirical Extended Tight-Binding (xTB). AL: data produced through active learning and NEB: transition states using Nudged elastic band theory (NEB); and based on stoichiometry - $SiO_n$: silicate clusters in a vacuum. $SiO_n\text{-}H_2O$: silicate clusters surrounded by water molecules, $H_2O$: water molecules only and with Na: molecular clusters of silicate or water with Na in it. **b** Two examples of attribution. The color bar is based on attributes. The red circles show the atomic environments added for active learning with atoms with high attributes at the center. The $H_3O^+ - H_2O$, $H_3O^+ - H_3O^+$, $H_3O^+ - OH^-$, $Si(OH)_4 - H_2O$ interactions and $Si(OH)_3OH_2^+$ complex lead to high attributes as they were not present in the training data. Parity plot of forces of a set of molecular clusters obtained through attribution-based active learning by **c** first generation NN-based inter-atomic potential (NNIP) and **d** final generation NNIP. The first-generation NNIP performs poorly on them while the final generation NNIP performs better. MAE: Mean absolute error. **e** Consecutive generation-wise improvement of attributes on a held-out condensed phase system. Attribution-based active learning thus improves our model. Source data are provided as a Source Data file.

executed at a computational cost much lower than first principle calculations. Several NNIP have been constructed for pure water[18–20], amorphous and liquid silica[21,22] and water-filled aluminosilicate zeolites[23]. However, no NNIP exists that models reactive silica-water interactions in acids or bases, which are required to understand the fundamental steps of silica precipitation, crystallization, and gel formation.

The initial step of silicate polymerization or oligomerization is dimer formation, whose mechanism depends on pH and temperature[24,25]. In neutral conditions or in a slightly acidic medium, the dimerization can occur through neutral monomers[24]. Aqueous reactions such as these are hard to understand at an atomic level, given the need to simulate the role of explicit solvent in the condensed phase, various bond breaking and formation, and proton-transfers in the solvent and reactants. Quantum chemical simulations can provide accurate energies for systems up to hundreds of atoms but are too costly to address free energy questions, that arise in the presence of full explicit solvation. A couple of possible molecular mechanisms have been reported for the neutral dimerization reaction[26]. One involves a $S_N2$-backside reaction[27] whereas the other involves a lateral flank-side attack[28].

In this study, we trained a reactive water-silica NNIP through an active learning loop. Our ground truth data consists of orbital-based hybrid DFT calculations on molecular clusters of various sizes. The trained NNIP generalizes to condensed phase simulation of thousands of atoms, both aqueous and solid silica. Since production simulations contain many more atoms than can be simulated with DFT we introduce an uncertainty attribution technique that builds on adversarial attacks on uncertainty[29] to locate clusters of uncertainty within large simulation boxes for annotating with DFT. The NNIP was validated

successfully against experimental properties of liquid water, and silica, both physical (diffusivity, structure, vibrational) and chemical (dissociation constants). We then used the potential to evaluate the dimerization of orthosilicic acid with over 2 ns of enhanced-sampling molecular dynamics (MD) simulations and resolve the previously unsettled mechanism in an aqueous solution.

## Results
### Reference Data Set
A preliminary training data set was constructed containing molecular silica clusters with explicit aqueous solvation. Molecular graphs for linear and cyclic $Si_xO_yH_z$ were generated with x <10 programmatically, and embedded into 3D conformers with RDkit[30]. Solvating water molecules were added at random with geometrical constraints to avoid clashes and to maximize hydrogen bonding. A subset of structures were deprotonated in one or more O-H bonds and calculated as anion, or balanced with $Na^+$ cations. We also added secondary and common building units of zeolites from the international zeolite database[31]. All structures were pre-relaxed with GFN2-xTB as implemented in xtb-6.4.1 and then refined with $\omega$B97X-D3/def2-TZVP level of theory in Orca[32], which served as our ground truth forces and energies. Reactive geometries for proton transfer and covalent Si-O-Si reactivity were obtained using a nudged elastic band at the GFN2-xTB level of theory with implicit solvent effect, followed by gradient DFT calculations at $\omega$B97X-D3/def2-TZVP. Figure 1a shows the distribution of the data set comprising 210K geometries based on sources as well as based on the stoichiometry of the molecule clusters.

Hybrid DFT is typically more accurate for bond-breaking than the GGA functionals used to train interatomic potentials on solids, but orbital-based simulations are restricted to a few hundred atoms in

molecule clusters. However, considering the local, atom-centered architecture of most NNIPs, they can learn from cluster data and run predictions on periodic systems. Most of the interactions in our concerning phase space are short-range coupled with electrostatic long-range interactions which are relatively shielded in a high dielectric medium like water[20,33].

## Active learning strategies

Our active learning methods rely on prediction uncertainty. An ensemble of three NNIP models was trained with varying parameters initialized with different random seeds and the variance of force predictions was used as the uncertainty metric. An uncertainty-based adversarial attack was applied by performing gradient-based optimization of the differentiable uncertainty metric[29]. New molecular conformations from the data set were sampled by back-propagating atomic displacements to find local optima that maximize the uncertainty of the NNIP committee while balancing thermodynamic likelihood. These new configurations were then evaluated using quantum mechanical calculations and used to retrain the NNIPs in an active learning loop.

## Differentiable uncertainty attribution

After early loops of adversarial attacks starting on the gen-0 dataset, the NNIP became stable enough to run MD simulations of condensed phases. An adversarial attack on condensed phases will only generate condensed phase configurations, while our data set comprises molecular clusters. Hence, we adapted the idea of attribution. Attribution is a technique employed to test how a change in a certain input neuron can impact an output neuron, thus enabling some interpretability[34,35]. Similarly, we calculated per-atom attribution of the force uncertainty, which can be qualitatively interpreted as which atoms contribute most to the uncertainty of the NNIP committee. Specifically, it is calculated as the derivative of the variance of forces or energy to atomic positions.

An active learning strategy around attribution allowed the selection of computationally tractable, high-uncertainty atomic clusters from MD frames of condensed phases without manual inspection. Atoms with attributes higher than two times the standard deviation from the mean are chosen as atomic centers and all neighboring molecules with at least one atom within a sphere of radius equal to the NNIP cutoff, the distance used to generate neighbors (6 Å), were chosen as the molecule cluster to add to training data. Hence, the approach collects the full first-solvation shell of any uncertain center, without disconnecting bonded atoms. The active learning approach is exemplified in Fig. 1b. The NNIP which is used to create the first generation of molecular clusters through attribution is tested on the same set of geometries after their DFT energies and forces were calculated. It performs very poorly while our current NNIP performs better as shown in Fig. 1c and Fig. 1d, thus proving the efficiency of attribution to choose atomic environments with high errors. Figure 1e shows a consecutive decrease in attributes over three generations showing that the attribution technique produces generalizable improvements to the potential, beyond the specific environments added in each generation. We also calculated attribution on bonds by taking a derivative to inter-atomic distances or bond lengths and showed it in Section 2 of the Supplementary document. Further, Supplementary Fig. 3 shows that without attribution-based active learning the base generation NNIP cannot simulate accurate structural properties of water.

## Properties of water

Benchmarking force error is not sufficient to quantify the quality of ML potentials[36] and simulation-based statistics should be used to evaluate model performance in production. We begin by testing the properties of liquid water.

## Radial distribution functions

We compared the oxygen-oxygen ($g_{OO}$) and oxygen-hydrogen ($g_{OH}$) radial distribution functions (RDFs) to experimental RDFs obtained from X-ray diffraction measurements[37] and those obtained by reported Gaussian moment neural network (GM-NN) potentials[20] trained on periodic systems of water as well as those trained on water clusters. Figure 2a shows good agreement of $g_{OO}$ with the experimental results with the first peak slightly overestimated similar to the ones obtained by BLYP and B3LYP potentials trained on clusters. Both the peak positions in $g_{OH}$ match experimental results as seen in Fig. 2b but are overestimated for all the NN potentials, which can be attributed to the lack of Quantum nuclear effects (QNE)[38] in the benchmark MD simulations. The O-O bond structure of water at ambient conditions is not as affected by QNEs as the bonds involving hydrogen[18,38] and we notice a similar behavior for our NNIP.

## Diffusion coefficient

The mean square displacements (MSDs) at different temperatures vs time are shown in Fig. 2c. At 300 K, diffusion coefficients (D) were calculated to be $0.21 \pm 0.005$ Å²/ps, close to an experimental value of $0.24 \pm 0.015$ Å²/ps[39]. Similar to RDFs, we compare our results to the GM-NN potentials whose predicted values are $0.182 \pm 0.006$ Å²/ps, $0.215 \pm 0.007$ Å²/ps, $0.143 \pm 0.004$ Å²/ps, $0.139 \pm 0.006$ Å²/ps respectively. We plotted our diffusion coefficients at different temperatures in Fig. 2d and compared our results with those obtained by another reported NN potential with MP2 accuracy (DP-MP2)[19] and experimental results[40]. Our results are higher than those predicted by the DP-MP2 model for most of the temperatures and are closer to experimental results. These underestimated D values for the DP-MP2 model are attributed to the overestimation of inter-molecular energy by MP2 with the aug-cc-pVDZ basis set. Thus our potential can perform better at predicting the diffusion properties of water at different temperatures and provide results comparable to experimental ones.

## Vibrational density of states

The vibrational density of states of water (VDOS) is shown in Fig. 2e, where we could observe that the peaks corresponding to O-H bond stretch, bending, and libration of water at 300 K obtained using our NNIP are quite close to the experimental observations[18]. We further employed Ring Polymer molecular dynamics (RPMD) coupled with path integral Langevin equation thermostat (PILE)[41] to introduce QNE. The RPMD simulations shifted the O-H stretch and bending peaks towards the left. The O-H stretch peak is shifted from 3670 to 3600 cm⁻¹ (3380 cm⁻¹ from the experiment) and the bending motion peak is shifted from 1664 to 1628 cm⁻¹ (1637 cm⁻¹ from the experiment). The libration has very minor changes due to QNE, also observed for VDOS obtained by DP-MP2 model[19] and in previous studies[18].

## Equilibrium density

Our NNIP predicted an equilibrium density of 1.08 g cc⁻¹. The GM-NN potentials produced liquid water with densities 0.86 g cc⁻¹, 1.02 g cc⁻¹, 1.1 g cc⁻¹, and 1.12 g cc⁻¹ respectively[20]. Our NNIP performed equally well as the GM-NN potentials trained on clusters. Considering our results we can claim that our NNIP though trained on molecular clusters can predict properties of bulk periodic water systems.

## Crystalline silica

We next tested our ability to predict the relative formation energies of crystalline silica like pure siliceous zeolites to take it a notch further and establish the success of our NNIP on periodic systems. We compared the relative formation energies of 236 zeolites to $\alpha$-Quartz predicted by our NNIP to the ones calculated using a periodic PBE-D3 level of DFT theory. We showed that our NNIP reproduces structural properties of $\alpha$-Quartz accurately in section 1 of the Supplementary document. Figure 3a depicts our NNIP predicting the relative energies

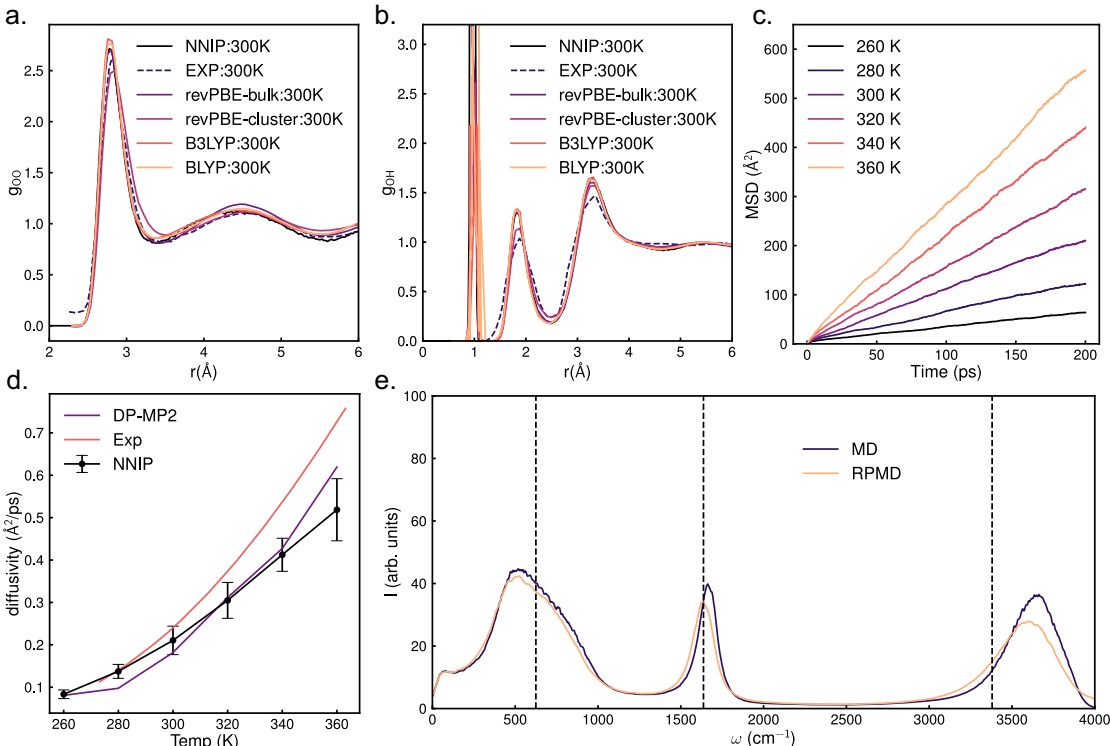

**Fig. 2 | Properties of water. a** O-O and **b** O-H radial distribution functions at 300 K were calculated by constant-temperature, constant-volume ensemble (NVT) simulations with our NNIP and compared to experimental and GM-NN potentials' results.NNIP: our NN-based inter-atomic potential; EXP: experimental data; revPBE-bulk: GM-NN potential trained on periodic water box calculated at the revPBE-D3 level of theory; revPBE-cluster: GM-NN potential trained on water clusters calculated at the revPBE-D3 level of theory; B3LYP: GM-NN potential trained on water clusters calculated at the B3LYP-D3 level of theory; BLYP: GM-NN potential trained on water clusters calculated at the BLYP-D3 level of theory. **c** Mean squared displacements vs time at different temperatures from 260 K to 360 K. **d** diffusivity vs temperature obtained using our NNIP compared with results from a DP-MP2 potential and experiment. The error bar shows the standard deviation of diffusivity prediction by our NNIP. **e** Vibrational density of states of liquid water obtained from classical molecular dynamics (MD) and Ring polymer molecular dynamics (RPMD) with our NNIP. The dashed vertical lines represent the experimental infrared spectrum. Source data are provided as a Source Data file.

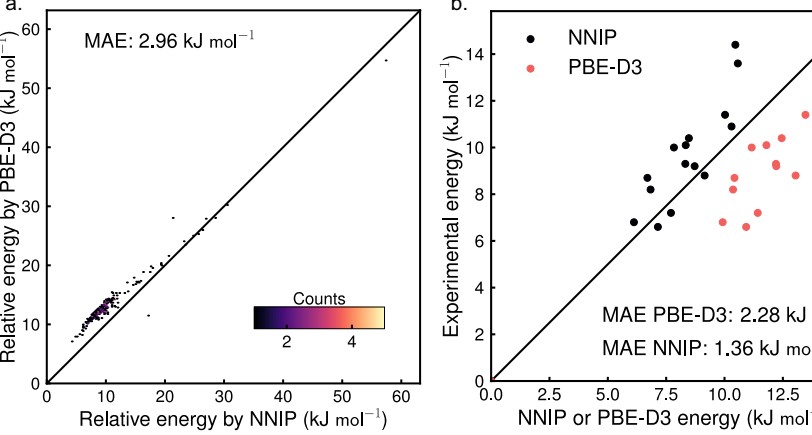

**Fig. 3 | Properties of crystalline silica. a** Relative formation energies of pure siliceous zeolites with respect to $\alpha$-Quartz calculated using our NN-based intera-tomic potential (NNIP) compared to those obtained by PBE-D3 level of Density Functional Theory (DFT). **b** Relative formation energies of fifteen zeolites obtained by NNIP and PBE-D3 compared to their experimental transition enthalpies. MAE Mean absolute error. Source data are provided as a Source Data file.

quite closely to periodic DFT results. We also collected experimental relative transition enthalpies for fifteen siliceous zeolites[42] and compared them to the relative formation energies calculated using periodic PBE-D3 DFT theory and our NNIP in Fig. 3b. The NNIP performed better than the periodic DFT. This better prediction can be because our NNIP is trained on data obtained using a higher level of hybrid functional DFT theory than PBE-D3. Thus, training on cluster data has a lesser impact than the theory of DFT employed to obtain the training

data and our NNIP serves better in predicting the relative formation energies of crystalline zeolites.

## Elastic constants

We also calculated elastic constants of $\alpha$-Quartz and siliceous FAU zeolite framework to further solidify our claim on NNIP's capability to predict properties of periodic systems involving silicates. The calculated elastic constants compared to results found in the literature are

**Table 1 | Elastic constants of α-Quartz and FAU siliceous zeolite framework calculated by our NNIP ensemble and compared to results published in the literature[66]**

| Crystalline silica | | Bulk Modulus (GPa) | Young Modulus (GPa) | Shear Modulus (GPa) | Poisson's ratio |
|---|---|---|---|---|---|
| α-Quartz | NNIP ensemble | 51.92 ± 10 | 112.16 ± 8 | 49.20 ± 3 | 0.14 ± 0.06 |
| | literature | 38.23 | 101.41 | 47.93 | 0.06 |
| FAU | NNIP ensemble | 44.34 ± 5 | 42.11 ± 4 | 15.69 ± 5 | 0.34 ± 0.02 |
| | literature | 61.37 | 50.24 | 18.42 | 0.36 |

The standard deviations obtained from the ensemble are also shown.

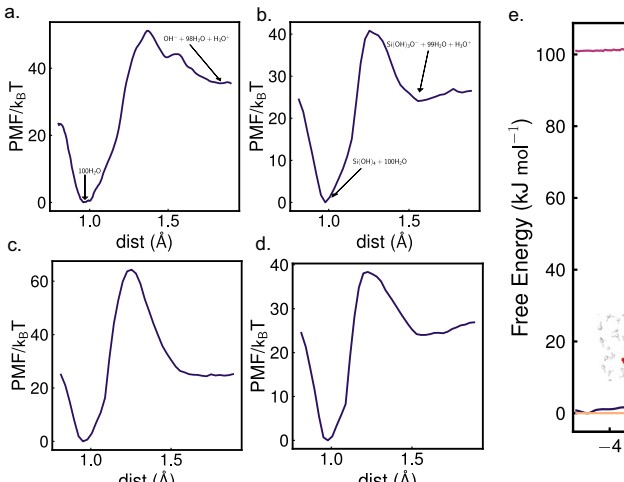
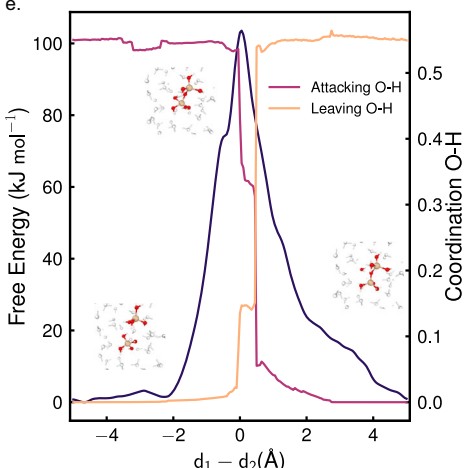

**Fig. 4 | Simulations of chemical reactions. a** Potential of mean force (PMF) of water deprotonation. The insets show reactant state (0.95 Å < $r_{OH}$ < 1 Å) and product state ($r_{OH}$ = 1.85 Å). The free energy of the reactant state is set to 0. **b** PMF of orthosilicic acid deprotonation. The insets show reactant state (0.95 Å < $r_{OH}$ < 1 Å) and product state ($r_{OH}$ = 1.55 Å). **c** PMF of dimer deprotonation. **d** PMF of trimer deprotonation. T is temp = 300 K. **e** Free energy profile of silica dimerization with lateral mechanism, coordination between attacking O and transferring hydrogen and that between leaving oxygen and the transferring hydrogen along the reaction coordinate $d_1 - d_2$ (Å). $d_2$ represents the bond distance between attacking oxygen and the attacked silicon, whereas $d_1$ represents the distance between the leaving oxygen and the attacked silicon. The insets show the reactant, product, and transition states. Source data are provided as a Source Data file.

shown in Table 1. It can be seen the NNIP ensemble is quite close to obtaining the elastic constants for both α-Quartz and FAU zeolite framework. Thus, our NNIP can capture the symmetry in periodic silicates adequately and can be used for predicting their properties.

## pK$_a$ of water

To obtain the pK$_a$ of water or pK$_w$, we chose a single H$_2$O molecule in a cubic box of 100 water molecules and varied one of the O-H bond lengths as our reaction coordinates from 0.8 to 1.9 Å with an increment of 0.05 Å. The deprotonation reaction is shown in equation (1)

$$H_2O(aq) + 99H_2O(l) \leftrightarrow OH^- + H_3O^+ + 98H_2O(l) \qquad (1)$$

Where H$_2$O (aq) is the water molecule undergoing dissociation. The potential of mean force (PMF) vs O-H bond length ($r_{OH}$) is shown in Fig. 4a. As seen from the figure, the PMF falls into a minimum between $r_{OH}$ = 0.95 and 1 Å. This is the equilibrium O-H bond length in the neutral water molecule, thus signifying the reactant state. We find another equilibrium point at $r_{OH}$ = 1.85 Å. Here, the H$^+$ has left the dissociated water molecule and has been successfully transferred to H$_3$O$^+$. The only interaction between OH$^-$ and the H$_3$O$^+$ is long-range Coulombic now and hence this state signifies the product state where the dissociation is complete. From the difference in PMF, we obtain our pK$_w$ as 15.4. The reference pK$_w$ value is 14[43] but considering the QNE the expected pK$_w$ becomes 17[44]. Our result is also comparable to a reported value of 16 obtained by first principle calculations using SCAN functional[45]. They used the same procedure as us to find the pK$_a$ of water.

## pK$_a$ of silicate oligomers

We also calculated the pK$_a$ of orthosilicic acid, silicate dimer, and trimer. Orthosilicic acid and dimer have all identical O-H bonds. Hence dissociating any one of them is fine. However, for trimer two O-H bonds attached to the middle Si are different than the end ones. We chose one of the end O-H bond dissociations for trimer to get its pK$_a$. Similar to water, we took a single orthosilicic acid, dimer, and trimer in a cubic box of 100 water molecules and varied their chosen O-H bond lengths. The individual PMFs vs $r_{OH}$ are shown in Fig. 4b–d. The deprotonation reaction of orthosilicic acid looks as shown in equation (2).

$$Si(OH)_4(aq) + 100H_2O(l) \leftrightarrow SiO_4H_3^- + H_3O^+ + 99H_2O(l) \qquad (2)$$

We found the equilibrium reactant state at $r_{OH}$ between 0.95 and 1 Å and product state at $r_{OH}$ = 1.55 Å for orthosilicic acid and between 1.5 and 1.55 Å for the dimer and the trimer. From the PMF, we obtain the pK$_a$ of orthosilicic acid as 10.45. Orthosilicic acid is a weak acid with a known pK$_a$ of 9.8[46]. Further, for dimer and trimer, we obtained a pK$_a$ of 10.43 and 10.42 respectively, whose pK$_a$ values have been reported to vary between 9.5 and 10.7[47]. Our reactive NNIP is thus shown to be capable of replicating complex reactions involving pure water as well as amorphous silicates in water.

**Fig. 5 | Schemes of silicate dimerization reactions. a** Expected scheme of $S_N2$ back side attack mechanism. The hydrogen from the attacking oxygen goes to the lateral oxygen, which then loses hydrogen to the leaving oxygen. **b** Observed scheme of back-side attack mechanism. When we chose the back side oxygen as leaving oxygen to replicate $S_N2$ mechanism, a surrounding water molecule interfered and took up hydrogen from the attacking oxygen forming a $H_3O^+$ and $OH^-$. **c** Observed lateral flank-side attack mechanism. This mechanism was not hampered by surrounding water molecules and occurred as reported in the literature. The transition state has a pentavalent Silicon atom.

## Silicate dimerization

We next simulated a dimerization reaction of two silicate monomers in a neutral solution with 100 water molecules at 300 K. We ran umbrella sampling on a reaction coordinate, $d_1 - d_2$ varying it from −5 Å to 5 Å with an increment of 0.05 Å. $d_2$ represents the bond distance between attacking oxygen and the attacked silicon, whereas $d_1$ represents the distance between the leaving oxygen and the attacked silicon. The attacking oxygen, attacked silicon, and the leaving oxygen atoms for the $S_N2$ and the lateral attack mechanisms are shown in Fig. 5. The $S_N2$ mechanism we observed differed from the predicted one in explicit solvent. We observed the formation of $H_3O^+$ and $OH^-$ which results in products with energy 79 kJ mol⁻¹ higher than the reactants and the activation energy is also higher (180 kJ mol⁻¹) than the lateral attack mechanism. The free energy profile of the $S_N2$ mechanism is shown in Supplementary Fig. 2. Hence, instead of the ions coming together to form the desired product, they tend to go back to the reaction state as shown in Fig. 5a–b. However, in the case of the lateral attack mechanism, the reaction in condensed solvent follows the proposed mechanism in continuum solvent[24–26], where the leaving oxygen leaves with the hydrogen from the attacking oxygen (Fig. 5c). This helps us to conclude that silicate dimerization in an aqueous solution at neutral conditions occurs through a lateral flank-side attack mechanism with pentavalent silicon as a transition state. Now from the free energy profile of the lateral mechanism in Fig. 4e, we see that at the coordinate close to 5 Å, the product dimer is formed. The reactant state with two monomers is at the coordinates between −3 and −5 Å where the reaction energy profile is almost flat. The product has 0.2 kJ mol⁻¹ more energy than the reactant phase. In a DFT study with an implicit solvent effect, the authors found the product has an energy 9 kJ mol⁻¹ higher than the reactant state with an activation energy of 129 kJ mol⁻¹ [24]. We obtained the activation energy as 103 kJ mol⁻¹ which can be trusted more as we used water solvent explicitly. From the plot of the coordination between the attacking oxygen with the transferring hydrogen and that of the leaving oxygen with the transferring hydrogen in Fig. 4e, we further observed that at the transition state, the hydrogen leaves the attacking oxygen with a drop in their coordination value and moves to the leaving oxygen with an increase in their coordination value almost immediately as expected in the predicted lateral flank-side mechanism.

## Discussion

Neural Network potentials have proven to work as a bridge between first principle calculations and classical potentials several times. In this work, we have proved the same by developing a reactive equivariant NNIP trained on a complex domain of silicate-water interactions. Our NNIP though trained on molecule clusters can predict properties of crystalline solids. The NNIP reproduces the physical and dynamic properties of water with high accuracy. We further proposed an active learning strategy based on attribution of differentiable uncertainty which further displays highly uncertain inter-atomic interactions in amorphous systems. This method not only serves as a qualitative measure for unsure atomic environments but also serves to minimize computational costs by extracting only the uncertain atomic environments. The reactive potential is further adept at predicting the reaction path of deprotonation of water, small silicate oligomers, and silica dimerization reaction in a water solution. This NNIP can further be used to probe into other silicate polymerization reactions leading to precipitated silica. We can also study the impact of pH and temperature on these reactions in the future as we have sodium in our data set to maintain charge neutrality or we can add aluminum in our data set to train a potential on preliminary stages of the synthesis of zeolites.

## Methods

### Model architecture

Our NN potential is based on PaiNN architecture[48], which is an equivariant neural network with message passing as its backbone. Equivariant models[48–50] can act on non-invariant inputs like displacement vectors in a symmetry-respecting way and thus offer good accuracy with a low amount of training data for properties that are equivariant in Euclidean space like forces, thus being more data sufficient than invariant potentials like Schnet[51]. In this method, an invariant feature vector for each atom is generated with their atomic numbers which are then updated through convolutions with "messages" from neighboring atoms which consist of distance, orientation, and features of nearest neighbor atoms within a cutoff distance. Each atomic feature is updated with information from neighbors and through convolutions obtains information from atoms far away as well, thereby generating a representation vector as a function of atomic positions and orientations, which is then mapped to atomic energies through feed-forward NN and summed to obtain the energy of the system. Through automatic differentiation, the forces on each atom can also be obtained. We further modified the model to predict stresses of a system based on the virial theorem[52] as given in equation (3) for pair potentials.

$$\sigma_{ij}^{V} = \frac{1}{V} \sum_{\alpha} \left[ \sum_{\substack{\beta=1 \\ \beta > \alpha}}^{N} \left( r_i^{\beta} - r_i^{\alpha} \right) f_j^{\alpha\beta} - m^{\alpha} v_i^{\alpha} v_j^{\alpha} \right], \quad f_j^{\alpha\beta} = \frac{\partial E}{\partial \left( r_j^{\beta} - r_j^{\alpha} \right)} \quad (3)$$

where (i,j) signifies x,y, and z directions, $\beta$ varies from 1 to N neighbors of atom $\alpha$, $r_i^{\beta}$, and $r_i^{\alpha}$ are positions of atom $\beta$ and $\alpha$ along direction i respectively, $f_j^{\alpha\beta}$ is the force on $\alpha$ by $\beta$ along direction j and E is total energy, V is total volume, $m^{\alpha}$ is the mass of atom $\alpha$ and $v^{\alpha}$ is the thermal velocity of atom $\alpha$.

### Training details

Molecular geometries of the same stoichiometry are split at a ratio of 3:1:1 into training, validation, and test sets. We employed a mini-batch gradient descent optimization with Adam optimizer. The learning drops from $10^{-4}$ to $10^{-6}$ at a rate of 0.5 when the validation loss hits a plateau for 20 epochs. The loss function is the weighted sum of square losses of forces and energies as implemented in PaiNN[48], with a loss coefficient of 0.95 for forces and 0.05 for energies. We fixed the number of convolutional layers at 4 and the cutoff for nearest neighbors at 6 Å. We optimized the other hyperparameters with SigOpt[53] on validation loss. The optimized dimension of the feature vector for each atom is 310; the number of radial basis functions used is 20 and the batch size is chosen as 9.

### Active learning

We ran adversarial attacks on molecular geometries randomly chosen from the available data set for three generations. Our model was now capable of running a few ps of simulations without exploding the system and we used it to run MD simulations on 10–100 orthosilicate molecules with ratios 1:2, 1:3, 1:5, and 1:10 of water molecules at 300, 500, 750, and 1000 K and ran three generations of attribution-based active learning on them. At high temperatures, we could sample silicate polymerization along with their intermediate and transition states. The fourth generation of attributions was run on simulations replicating the deprotonation of water, orthosilicic acid, and other silicates. We then ran one generation of attributions on zeolite crystals and the sixth generation on the neutral dimerization reaction to extract uncertain molecule clusters from them. Our active learning thus consists of a total of nine rounds, involving three from adversarial attacks and then six generations of attribution-based active learning on the various stoichiometries.

### Simulation details

We used the atomic simulation environment (ASE) package coupled with our NNIP to run simulations. All periodic structures with silicates and water molecules are generated using Packmol[54] and rendered using OVITO[55]. The simulation boxes were generated with the equilibrium density of water predicted by our NNIP of 1.08 g cc$^{-1}$. All MD simulations were run with canonical (NVT) ensemble with Nosé-Hoover thermostat and timestep of 0.5 fs.

### Radial distribution functions of water

We obtain the oxygen-oxygen ($g_{OO}$) and oxygen-hydrogen ($g_{OH}$) RDFs from 100 ps long MD simulations of a periodic box containing 400 $H_2O$ molecules in canonical (NVT) ensemble at 300 K after 20 ps equilibration period. We divide the whole trajectory into ten parts and take the average RDF curve as the estimated one.

### Diffusion coefficients of water

The diffusion coefficients were calculated from the MSDs by Einstein's diffusion equation as given in equation (4)

$$D = \frac{MSD}{6\Delta t} = \frac{\langle |x(t + \Delta t) - x(t)|^2 \rangle}{6\Delta t} \quad (4)$$

Where D is the diffusion coefficient, $\Delta t$ is the time interval used for measuring MSDs. We used 10 ps as the time interval. We further obtained the slope of ln(MSD) vs ln(t) and choose the region for linear fitting where the slope is within $1 \pm 10^{-4}$. The MSDs were calculated based on the relative squared displacement of oxygen atoms and averaged over trajectories for all water molecules. We further needed to extrapolate the diffusion coefficient to infinite-system size owing to the finite-size effect[56]. The extrapolated diffusion coefficient (D(∞)) is obtained from finite-system size diffusion coefficient (D(L)) with an extra correction term as shown in equation (5)

$$D(\infty) = D(L) + \frac{\zeta}{6\pi\beta\eta L} \quad (5)$$

Where L is the simulation box size, $\zeta$ depends on the geometry of the simulation box (2.837297 for a cubic box), $\beta$ is $\frac{1}{K_B T}$, and $\eta$ is experimental shear viscosity = 0.8925 x $10^{-3}$ Pa s. We calculated the diffusion coefficients by first running an NVT simulation for 50 ps to equilibrate the system and then ran five independent NVE simulations for 150 ps at different temperatures of 260, 280, 300, 320, 340, and 360 K and chose the average MSDs.

### Vibrational density of states of water

The VDOS of water can be procured from the Fourier transform of the velocity-velocity autocorrelation function as given in equation (6)

$$C_{vv}(\omega) = \int \langle v(\tau)v(\tau + t)_\tau \rangle e^{-i\omega t} dt \quad (6)$$

Where v($\tau$) is the centroid velocity of the atom and $\omega$ is the vibrational frequency. The RPMD simulations were implemented using SchnetPack[57] with 5 beads and 0.25fs timestep.

### Equilibrium density of water

We ran MD simulations in NPT (constant pressure) ensemble using the isobaric-isothermal form of the Nosé-Hoover dynamics at a temperature of 300 K and a pressure of 1 atm. We chose a cubic box of 400 water molecules and the box was allowed to relax in a hydrostatic manner. A 1 ns long trajectory after 100 ps of equilibration period was divided into ten parts and the time-averaged volume of the simulation box was taken as equilibrium volume. The equilibrium density was then calculated from it.

## Elastic constants of silicates

We provided normal strains of −1 to 1% in steps of 0.5% and shear strains of −4 to 4% in steps of 2% to the equilibrium structures of $\alpha$-Quartz and FAU zeolite and obtained their deformed structures using Pymatgen[58]. We then optimized each deformed structure by the BFGS algorithm implemented in the ASE package. We then obtained the stresses at different strains and their corresponding slopes to measure the elastic tensors for each system. We then used ELATE software[59] to obtain elastic constants from the tensors. We used our NNIP ensemble to procure the average elastic constants and their standard deviation.

## Chemical reactions

We performed constrained MD simulations with umbrella sampling varying our reaction coordinates chosen for different reactions. We then obtained a smooth reaction path from reactant to product state by calculating the PMF as a function of our chosen reaction coordinates. The PMF was calculated using the Multistate Bannett's Acceptance Ratio (MBAR)[60] on the sampled trajectories. The MBAR analysis was carried out using codes implemented in adaptive sampling package[61,62]. We can use the PMF of the deprotonation reactions of water and silicates to obtain their $pK_a$ values. The difference in the PMF of reactant and product phase ($\Delta F$) is related to the acid dissociation constant ($K_c$) as shown in equation (7) and the relation between $pK_a$ and $K_c$ is shown in equation (8), Combining both we can get $pK_a$ from $\Delta F$ as shown in equation (9).

$$\Delta F = -k_B T \ln k_c \tag{7}$$

$$pK_a = -\log k_c \tag{8}$$

$$pK_a = \frac{\Delta F}{k_B T \times 2.3026} \tag{9}$$

Where $k_B$ is the Boltzmann constant and T is temp = 300 K.

## Data availability

The silica-water data set and the NNIP models generated during this study are deposited at Materials Data Facility[63] at https://doi.org/10.18126/PZJR-X7PV. Source files for all graphs in our figures are provided in the Source Data folder deposited at figshare[64].

## Code availability

The codes used for this study can be downloaded from the following github repository: https://github.com/learningmatter-mit/NeuralForceField under the MIT License (see ref. 65 for permanent link).

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

## Acknowledgements

R.G-B. and S.R. acknowledge funding support from Evonik AG and the MIT-IBM Watson AI Lab. We acknowledge the MIT Engaging cluster at the Massachusetts Green High-Performance Computing Center (MGHPCC) for providing high-performance computing resources.

## Author contributions

R.G-B. conceived the project. S.R. designed methodologies, curated data, and implemented computer codes and algorithms. J.P.D. contributed to programming. S.R. analyzed the results. R.G-B., J.P.D., and T.S.A. contributed to the analysis and discussion. F.Z. provided some initial datasets.

R.G-B. supervised the research. S.R. wrote the first draft of the manuscript and J.P.D., T.S.A., and R.G-B. contributed to the final version.

## Competing interests

The authors declare no competing interests.
