## [Peer Review File · Nature Communications]

Learning a reactive potential for silica-water through uncertainty attributionREVIEWER COMMENTS

Reviewer #1 (Remarks to the Author):

This paper presents a study where the reactivity of silicates in solution with water is investigated using an existing neural network based potential. The reference datasets used for training have been curated using an previously proposed active learning approach that estimates model uncertainties from an ensemble of three independently trained potentials.

The main objective of the paper, which is to "demonstrate how a machine-learning reactive interatomic potential can accurately capture silicate-water reactivity" is achieved: the generation of the dataset is well-documented and several properties and observables for pure water, crystalline silica and amorphous silicates in water are successfully calculated using the resulting model. These analyses include radial distributions functions for water, diffusion coefficients, vibrational density of states, equilibrium densities, pKa's and dimerization reactions of silicate monomers in water.

In the conclusion, the authors claim to have "developed a reactive equivariant NNIP trained on a complex domain of silicate-water interactions", but it is unclear from the manuscript, what this development entails exactly.

From the details provided in the paper, the proposed NNIP model is based on the existing PaiNN potential, supplemented by an extra output module to predict stresses (as shown in Eq.3). The active learning framework used in this study has also been previously proposed by the same authors. It is inconclusive, what modifications were done to the original PaiNN architecture and the active learning algorithm in order to enable the presented study. Is it just a simple modification of the parameters or does it include functional changes? What aspects were missing from the original formulation of PaiNN and how do these changes improved the results for the presented study?

Overall, the paper is well written, with good organization and structure. The narrative is cohesive and presented in clear and concise language. It is easy to follow the line of reasoning. However, this paper does not seem to represent a methodological advance and I am not convinced that it suits the interdisciplinary readership of Nature Communications.

Reviewer #2 (Remarks to the Author):

This manuscript is well organized and contributes to advancing silicate research. The authors use state-of-the-art neural network inter atomic potentials to study the reactivity of silicate in water. This paper is suitable to be published in the Nat Comm, although I have a few minor observations that the authors could use to improve the paper:

- There is some ambiguity in how the main DFT method to create the data is specified, for example, it is written in two places ' ω -B97XD\def2-TVZP' and ' ω -B97xD3/TZVP', but it should be ' ω B97x-D3/def2-TZVP'.
- The DFT software ORCA should be cited as required in the license.
- The 'xTB' deserves a full citation and the text should include the parameters used.
- In the section '2.4.1 Elastic constants' the elastic constants of alpha-quartz and FAU are not reported in reference 40 (or I am missing them), or it is not clear to me whether the 'literature' values in Table 1 come from reference 40. This could be explicit in the captions of the Table.
- Half of the paper is dedicated to the extensive validation and discussion of the water NNIP, this should be reflected in the abstract and conclusions.

Reviewer #3 (Remarks to the Author):

The authors present a new parameterization of a neural network-based interatomic potential (NNIP), trained through an active learning approach, where an "attribution" measure is used to identify configurations of molecular clusters (i.e., atomic configurations) necessary to add to training data for improved accuracy. This training procedure was applied to the parameterization of a water/silica material system. Diffusivity, radial distributions, vibrational, and chemical processes were demonstrated for the NNIP.

Novel parameterizations of machine learned interatomic potentials (ML-IAP) are important to the research communities using classical molecular dynamics (MD). If trained correctly, ML-IAPs can

provide 1) an accuracy close to that of density functional theory (DFT), 2) the capability to model more representative atomic configurations with this accuracy (i.e., number of atoms significantly greater than DFT, > millions), and 3) longer durations of simulation time (greater than that of DFT, > ps). Material interactions in silica-water systems are still a highly active area of research, where furthering a molecular-scale understanding of dynamic and chemical processes can make a significant scientific impact. This makes the overall topic of the manuscript valuable to the materials science community.

The manuscript was difficult to evaluate, and I found myself somewhat confused about its intention, because it has traits of both methods and materials. I feel these two aspects to the story of the manuscript are both incomplete. The first aspect is the method of developing the NNIP. There is little to no discussion or demonstration about how the NNIP performed relative to key parameters. For example, as a user of the proposed parameterization method, I would like to see further analysis of behaviors like convergence based on the generation of active learning or depth of the NN, how those parameters impact accuracy, and how they were chosen, along with training/testing relationships. A key question I had was, what considerations are necessary for later generations of the active learning process when there are no strong outliers? Notably, this part of the story was only a small contribution to the manuscript in terms of area. The second aspect of the story is how the NNIP performs when modeling silica/water. The manuscript leaves any practitioner of MD on their own to test several key properties before trusting its accuracy. I believe further tests should be included for completeness. For example, simple structural metrics are missing and, if present, could provide further evidence for validation. These include well-known, computationally, and experimentally validated quantities like bond length and angular distributions, along with their temperature dependence. It is argued in the manuscript that although trained on small molecular clusters, the NNIP is applicable when modeling condensed phases of silica. I feel the authors have missed the mark in convincing the reader of the accuracy and transferability here, and further analysis would be appropriate. As an example, a quantification of structural properties in crystalline quartz as predicted by NNIP should be provided and compared to DFT and experiment.

In the current state, I would suggest rejecting the manuscript. I would reconsider with major revisions to both aspects of the manuscript's story. A few detailed concerns follow.

- Line 105-106, I interpreted the strategy of selecting configurations for addition to the training data to be a critical part of this manuscript's effort. What is meant by "balancing" the likelihood? To what degree could these "back-propagated" configurations be non-physical? Other than high attribution, is there any metric to suggest that selected configurations are the "best" or "most effective" configurations to choose for training improvement? Are these new configurations transition states, equilibrium states, or metastable? Are these configurations then also "relaxed" (line 86)? If so, are they unique to the total data set? Or does the relaxed configuration possibly repeat existing configurations? Please provide more detail here.
- Line 126 and Figure 1b: It is very difficult to identify from the figure what these configurations show even though they are labeled.
- Figure 1c, d, e: It is agreed that the parity plots of force indicate that a subsequent generation of the active learning process improves results. However according to the caption, (d) is showing data from the "current" generation and there are still significant outliers. What does "current" mean? Additionally, in Figure 1e, the outliers in attribution seem to diminish or disappear almost completely. Will this active learning approach continue to improve the results of figure 1d with increased number of generations, and how does the reader know this? When do further generations stop providing improvements to results?
- Line 159: Why are the authors not comparing the NNIP temperature dependence of D to experiment? Instead, they introduce a new NN model and compare to it. These values for water are well-studied experimentally and would tell the reader a lot about the accuracy and what to expect from the NNIP.
- Subsection 2.3.3: This paragraph seems to be focused about results from an RPMD model and not the NNIP. Again, a new model is introduced, why? Is RPMD known to be particularly accurate in the vibrational spectra of water due to NQE? How does the reader know that both results are not equally incorrect? Is it possible for the authors to compare NNIP results to experiment, maybe through a spectroscopic method? In fact, a reader would find more value in a comparison between

their DFT results, experiment, and NNIP, rather than only making comparisons to yet another model.

- Section 2.3.4: The NNIP predicts bulk densities of water greater than known from experiment. Can the authors further explain this and the relationship to the cutoff for the NNIP being 6Å? Here, any molecular cluster used for training should include multiple water molecules. If this is true, shouldn't the NNIP show better accuracy in density? What does the distribution of attribution look like? Is attribution still a strong indicator of atom centers necessary for active learning. How is attribution and hydrogen-bond length related? Please explain this predicted high density.
- Section 2.4: In figure 3a the NNIP is shown to under-predict the relative energy of nearly all 236 zeolites in relation to PBE-D3. In line 188, there is a claim that NNIP performed better than DFT and was demonstrated for 15 specific zeolites in Figure 3b. This characteristic was later attributed to the NNIP being trained on a higher level of hybrid functional. To back up this claim, the authors could calculate the remainder of the 236 zeolites using a different (i.e., better, or more appropriate) hybrid functional and reproduce figure 3a to convince the reader.
- Figure 3: Formation energies were compared relative to that of alpha-quartz. I assume the NNIP results were normalized by NNIP alpha-quartz and the PBE-D3 results were normalized by PBE-D3 alpha-quartz. If true, how does the reader know that the NNIP gets the energetics, structure, forces, angles, bond lengths, etc. of quartz correct? By comparing relative energies, are the data being obscured or misrepresented?
- Line 250-255: The sentence is difficult to understand and interpret at best. I'm confused as to how this makes the point of which mechanism occurs here. Maybe clarity could come from labels of attacking, leaving, etc. in Figure 5 and significant rewording.
- Atomic charge can play a key role in the dynamics of water and water/silica interactions. The NNIP does not include charge to the best of my understanding. Notably, the authors chose to address the acidic (low pH) chemical pathway for dimerization occurring through neutral reactant species. In the context of NNIP transferability, a discussion of the role of charge and the implications of NNIP not including charge is warranted. If NNIP does include charge, I missed that fact, and a discussion of how charge is included is warranted.

Others

Lines 135: Incomplete sentence, wording?

Missing citations: Line 81: "RDKit", Line 86: "Orca", line 319 "Packmol", line 319: "Ovito", line 358: "Pymatgen"

Line 168, 172: "...NQE..." do you mean QNE as abbreviated earlier?

Reply to Reviewers

Reviewer 1

Comments

This paper presents a study where the reactivity of silicates in solution with water is investigated using an existing neural network-based potential. The reference datasets used for training have been curated using a previously proposed active learning approach that estimates model uncertainties from an ensemble of three independently trained potentials.

Authors - We greatly appreciate the reviewer for going through our paper and for providing deep insights. Your feedback is deeply appreciated. Our reference dataset was obtained using not just the previously proposed adversarial attack, but also a novel method of attribution-based active learning as detailed in section 2.2.1. Differential uncertainty attribution.

The main objective of the paper, which is to “demonstrate how a machine-learning reactive interatomic potential can accurately capture silicate-water reactivity” is achieved: the generation of the dataset is well-documented and several properties and observables for pure water, crystalline silica, and amorphous silicates in water are successfully calculated using the resulting model. These analyses include radial distribution functions for water, diffusion coefficients, vibrational density of states, equilibrium densities, pKa’s, and dimerization reactions of silicate monomers in water.

In the conclusion, the authors claim to have “developed a reactive equivariant NNIP trained on a complex domain of silicate-water interactions”, but it is unclear from the manuscript, what this development entails exactly. From the

details provided in the paper, the proposed NNIP model is based on the existing PaiNN potential, supplemented by an extra output module to predict stresses (as shown in Eq.3). The active learning framework used in this study has also been previously proposed by the same authors. It is inconclusive, what modifications were done to the original PaiNN architecture and the active learning algorithm in order to enable the presented study. Is it just a simple modification of the parameters or does it include functional changes? What aspects were missing from the original formulation of PaiNN and how do these changes improve the results for the presented study?

Authors - We did not add significant modifications to PaiNN architecture. The original formulation already has stress implemented. Here, we merely added the feature of obtaining virial stress which is not contributing to any improvement in results. However, our paper does not claim to have modified PaiNN, rather we used the architecture to develop a reactive equivariant NNIP as the reviewer pointed out which can begin to decipher the complex domain of silicate-water interactions starting with silicate dimerization reaction at a neutral medium in explicit water solvent as we have shown in our results. Further, along with the previously introduced active learning method, we have also introduced an attribution-based active learning method that helps us train our model on molecular clusters but run simulations on periodic systems. In this paper, we have introduced a novel active learning method and a reactive NNIP that can resolve the silicate dimerization reaction which is a primary step in silicate polymerization and zeolite formation.

Overall, the paper is well written, with good organization and structure. The narrative is cohesive and presented in clear and concise language. It is easy to follow the line of reasoning. However, this paper does not seem to represent a methodological advance and I am not convinced that it suits the interdisciplinary readership of Nature Communications.

Authors – In our work, we have presented two advances, one scientific and another methodological. We have created a reactive NNIP that can transverse well in the complex domain of silica-water interactions. Further, we were able to use it to resolve the silica dimerization reaction in a neutral medium with explicit water solvent. This NNIP has now potential to

further probe into the silica precipitation and zeolite formation stages. Secondly, we have proposed a new active learning method based on per-atom uncertainty attribution that helps train molecular clusters whose forces and energies can be calculated at a higher level of hybrid functional theory and then run simulations on big periodic systems.

Reviewer 2

Comments

This manuscript is well organized and contributes to advancing silicate research. The authors use state-of-the-art neural network inter-atomic potentials to study the reactivity of silicate in water.

This paper is suitable to be published in the Nat Comm, although I have a few minor observations that the authors could use to improve the paper:

1. There is some ambiguity in how the main DFT method to create the data is specified, for example, it is written in two places ' ω -B97XD/def2-TVZP' and ' ω -B97xD3/TZVP', but it should be ' ω -B97x-D3/def2-TZVP'.

Authors – We agree with the reviewer. It indeed will be ' ω -B97x-D3/def2-TZVP'.

Action taken – We have changed line 86 to ω -B97x-D3/def2-TZVP.

2. The DFT software ORCA should be cited as required in the license.

Action taken – We have added a reference.

3. The 'xTB' deserves a full citation and the text should include the parameters used.

Authors – We have added references. We have used GFN2-xTB for optimization and GFN2-xTB with implicit solvent effect for NEB transition states.

Actions taken – We modified the lines 86-90. All structures were pre-relaxed with GFN2-xTB as implemented in xtb-6.4.1 and then refined with ω -B97xD3/def2-TZVP level of theory in Orca, which served as our ground truth forces and energies. Reactive geometries for proton transfer and covalent Si-O-Si reactivity were obtained using a nudged elastic band at the

GFN2-xTB level of theory with implicit solvent effect, followed by gradient DFT calculations at ω -B97xD3/def2-TZVP.

4. In the section '2.4.1 Elastic constants' the elastic constants of alpha-quartz and FAU are not reported in reference 40 (or I am missing them), or it is not clear to me whether the 'literature' values in Table 1 come from reference 40. This could be explicit in the captions of the Table.

Authors – We apologize for the confusion and will add the references in the caption.

Action taken – References are added in Table 1 caption.

5. Half of the paper is dedicated to the extensive validation and discussion of the water NNIP, this should be reflected in the abstract and conclusions.

Authors – We modified our conclusion accordingly.

Reviewer 3

Comments

The authors present a new parameterization of a neural network-based interatomic potential (NNIP), trained through an active learning approach, where an “attribution” measure is used to identify configurations of molecular clusters (i.e., atomic configurations) necessary to add to training data for improved accuracy. This training procedure was applied to the parameterization of a water/silica material system. Diffusivity, radial distributions, vibrational, and chemical processes were demonstrated for the NNIP.

Novel parameterizations of machine-learned interatomic potentials (ML-IAP) are important to the research communities using classical molecular dynamics (MD). If trained correctly, ML-IAPs can provide 1) an accuracy close to that of density functional theory (DFT), 2) the capability to model more representative atomic configurations with this accuracy (i.e., number of atoms significantly greater than DFT, > millions), and 3) longer durations of simulation

time (greater than that of DFT, $>$ ps). Material interactions in silica-water systems are still a highly active area of research, where furthering a molecular-scale understanding of dynamic and chemical processes can make a significant scientific impact. This makes the overall topic of the manuscript valuable to the materials science community.

Authors - We would like to extend our sincere thanks to the reviewer for recognizing and highlighting the significance of our paper in the reviewer's feedback.

The manuscript was difficult to evaluate, and I found myself somewhat confused about its intention, because it has traits of both methods and materials. I feel these two aspects to the story of the manuscript are both incomplete. The first aspect is the method of developing the NNIP. There is little to no discussion or demonstration about how the NNIP performed relative to key parameters. For example, as a user of the proposed parameterization method, I would like to see further analysis of behaviors like convergence based on the generation of active learning or depth of the NN, how those parameters impact accuracy, and how they were chosen, along with training/testing relationships. A key question I had was, what considerations are necessary for later generations of the active learning process when there are no strong outliers? Notably, this part of the story was only a small contribution to the manuscript in terms of area. The second aspect of the story is how the NNIP performs when modeling silica/water. The manuscript leaves any practitioner of MD on their own to test several key properties before trusting its accuracy. I believe further tests should be included for completeness. For example, simple structural metrics are missing and, if present, could provide further evidence for validation. These include well-known, computationally, and experimentally validated quantities like bond length and angular distributions, along with their temperature dependence. It is argued in the manuscript that although trained on small molecular clusters, the NNIP is applicable when modeling condensed phases of silica. I feel the authors have missed the mark in convincing the reader of the accuracy and transferability here, and further analysis would be appropriate. As an example, a quantifica-

tion of structural properties in crystalline quartz as predicted by NNIP should be provided and compared to DFT and experiment.

Authors - We have tried to answer the reviewer’s detailed concerns to make the manuscript clearer. The hyperparameters are chosen through a Bayesian optimization framework in SigOpt (<https://sigopt.com/>) as described in Methods section 4.2 Training details. The number of convolutional layers and cutoff were chosen as 4 and 6 Å before the hyperparameter optimization loops. More convolutions mean more farther-out neighbors are implicitly included in the representation of the chemical environment in a message-passing network. However, a large number of convolutional layers can lead to a slower network with only slight improvement in predictions, so we chose 4. We have answered the question regarding later generations of active learning in detail below. Our goal with this potential is to replicate the silicate polymerization reactions in water at different environmental conditions. We tested our model’s potential by predicting some of the structural and dynamic properties as well as reactions. We agree that providing further experimentally validated quantities can further validate our model but we feel the results provided are already enough to trust our model for the silicate dimerization reactions.

Actions taken - We have calculated the structural properties of crystalline quartz and have compared it to DFT and experiment and mentioned it later as an answer to a more detailed question.

1. Lines 105-106, I interpreted the strategy of selecting configurations for addition to the training data to be a critical part of this manuscript’s effort. What is meant by “balancing” the likelihood? To what degree could these “back-propagated” configurations be non-physical? Other than high attribution, is there any metric to suggest that selected configurations are the “best” or “most effective” configurations to choose for training improvement? Are these new configurations transition states, equilibrium states, or metastable? Are these configurations then also “relaxed” (line 86)? If so, are they unique to the total data set? Or does the relaxed configuration possibly repeat existing configurations? Please provide more detail here.

Authors - In active learning method based on adversarial attack, not only uncertainty is

maximized, but rather the product of Boltzmann likelihood and uncertainty, to sample configurations that are likely to be observed during production simulations, but also far from the training data. More details on adversarial attacks can be found in the paper it was introduced [1].

This way, the uncertainty of the samples is balanced with their likelihood of restricting our search to representative configurations. However, continually increasing uncertainty may outbalance the likelihood term and reach non-physical geometries that would never be visited during production simulations, but it is straightforward to stop sampling before that is the case. Further details are mentioned in the original paper [1].

As regards, attribution-based active learning, we choose atom centers with high attributed uncertainty and, to create fully connected chemical patterns, the configurations chosen for DFT labeling are the full set of atoms within a 6 sphere centered on the high-attribution atom, as well as any additional molecular fragments outside the sphere but covalently connected to some atom within the sphere.

We ran DFT on these configurations and found that the potential performs poorly on them as shown in Fig 1c, as expected from the high uncertainty attribution. Thus, adding these configurations to the training pools improves the model, and the attribution strategy was able to single out these local uncertainty environments.

Our potential is used to explore structural, dynamic, and reactive simulations including water properties, silicate crystal properties, and then deprotonation and dimerization reactions. We ran consecutive attribution-based sampling on all these simulations until no high-attribution centers were detected during production simulations. This methodology is explained in section 4.3. Active learning. These isolated configurations were added to training data irrespective of them being in transition, metastable, or equilibrium states. The configurations in the preliminary dataset are only relaxed as mentioned in 2.1. Reference Data set.

After each batch of attribution, we compared the chosen geometries and rejected ones that were very close to the environment already selected for new DFT calculations, to avoid duplicated simulations on essentially identical geometries. More details are given in the Methods section 4.3 Active learning.

Actions taken- we modified the section 4.3.

We ran adversarial attacks on molecular geometries randomly chosen from the available data set for three generations. Our model was now capable of running a few ps of simulations without exploding the system and we used it to run MD simulations on 10-100 orthosilicate molecules with ratios 1:2, 1:3, 1:5, and 1:10 of water molecules at 300, 500, 750, and 1000 K and ran three generations of attribution-based active learning on them. At high temperatures, we could sample silicate polymerization along with their intermediate and transition states. The fourth generation of attributions was run on simulations replicating the deprotonation of water, orthosilicic acid, and other silicates. We then ran one generation of attributions on zeolite crystals and the sixth generation on the neutral dimerization reaction to extract uncertain molecule clusters from them. Our active learning thus consists of a total of nine rounds, involving three from adversarial attacks and then six generations of attribution-based active learning on the various stoichiometries.

2. Line 126 and Figure 1b: It is very difficult to identify from the figure what these configurations show even though they are labeled.

Actions taken - We have replaced Figure 1b with a higher resolution version in which the configurations are more visible.

3. Figure 1c, d, e: It is agreed that the parity plots of force indicate that a subsequent generation of the active learning process improves results. However according to the caption, (d) is showing data from the “current” generation and there are still significant outliers. What does “current” mean? Additionally, in Figure 1e, the outliers in attribution seem to diminish or disappear almost completely. Will this active learning approach continue to improve the results of figure 1d with increased number of generations, and how does the reader know this? When do further generations stop providing improvements to results?

Authors - The “current” generation means here the finalized NNIP that we used to obtain our various results and hence is renamed as “final” generation in the caption.

Even with outliers, the final generation of NNIP shows accuracy in forces with an MAE of 1.32 kcal/mol. As we mentioned in section 2.3, accuracy in forces is not enough, the potential needs to predict properties of the system comparable to established results [2]. Here, our

approach is the same. As mentioned in the Methods section 4.3, we ran attribution on different configurations of different stoichiometries related to reactions or properties we aimed to reproduce. Figure 1e shows that the relative uncertainty on each atom in the silicate cluster-water configurations decreased almost completely, depicting our potential’s ability to interpolate silicate-water interactions. Our stopping criteria for active learning do not consider the accuracy of force predictions or per-atom attributions but the overall behavior and outcomes of the simulations run by our model. It implies that the model should reach a point where it provides reliable and consistent results comparable to those observed in experiments or documented in the literature. Figure 1e also shows that with increasing data the model improves and thus attribution which maps the uncertainty also decreases. More data also would improve force errors and the results of Figure 1d. Thus, continuing this active learning approach will probably lead to better force accuracy.

Further generations will keep on providing improvements until the number of molecular clusters gets exhausted or gets too low to exert an impact on the model’s performance.

Actions taken – We changed the caption of Figure 1d.

4. Line 159: Why are the authors not comparing the NNIP temperature dependence of D to experiment? Instead, they introduce a new NN model and compare to it. These values for water are well-studied experimentally and would tell the reader a lot about the accuracy and what to expect from the NNIP.

Authors - We were comparing our diffusion coefficients to a reported NN potential with MP2 accuracy (DP-MP2) [3]. However, we took the reviewer’s advice and compared our results to experimental values as well. Our room temperature diffusivity is close to experimental data at lower temperatures but as temperature increases, our activation energy for diffusion keeps increasing but qualitatively our predicted diffusivity shows correct behavior.

Actions taken – We plotted the experimental values and compared them to our model in Figure 2d.

5. Subsection 2.3.3: This paragraph seems to be focused about results from an RPMD model and not the NNIP. Again, a new model is introduced, why? Is RPMD known to be particularly accurate in the vibrational spectra of water due to NQE? How does the reader know that both results are not equally incorrect?

Is it possible for the authors to compare NNIP results to experiment, maybe through a spectroscopic method? In fact, a reader would find more value in a comparison between their DFT results, experiment, and NNIP, rather than only making comparisons to yet another model.

Authors - Ring Polymer Molecular Dynamics is a simulation method that involves a classical approach in an extended Ring-polymer phase space to approximate the QNE. We use the same NNIP as our potential here. The QNE has previously been shown to have an impact on the vibrational spectra of water [3, 4].

Our results from both MD simulations and RPMD simulations with our NNIP are indeed compared with experimental infrared spectrum peaks as shown and mentioned in Figure 2e.

6. Section 2.3.4: The NNIP predicts bulk densities of water greater than known from experiment. Can the authors further explain this and the relationship to the cutoff for the NNIP being 6Å? Here, any molecular cluster used for training should include multiple water molecules. If this is true, shouldn't the NNIP show better accuracy in density? What does the distribution of attribution look like? Is attribution still a strong indicator of atom centers necessary for active learning. How is attribution and hydrogen-bond length related? Please explain this predicted high density.

Authors - The NNIP does have multiple water molecules in clusters used for training. The 6 Å is the cutoff for first-order interactions in message passing. Then all the atoms within this cutoff exchange their local information, meaning that a central atom now has “information” about atoms 12 Å away, and for each convolution layer, it acquires information from atoms 6 Å away. Thus, our NNIP having four convolutional layers gathers information on atomic environments of 24 Å, which is larger than our molecular clusters in the training data. Thus the model learned some “vacuum signature”. This does pose a challenge for the NNIP to infer the local fingerprint of condensed phase environments. Even though we have found our model surprisingly capable of reproducing condensed phase properties from these cluster data, the new “infinitely solvated” chemical environments are likely somewhat different from the “cluster” chemical environments in a way where our model somewhat overbinds. In addition to this, different DFT functionals have been shown to predict different

densities of water at ambient conditions. The density of liquid water with SCAN functional was predicted higher than the experimental value (1.05 g/cc) [5]. Further, a study on DFT with a dispersionless generalized gradient approximation functional required higher pressure and temperature than ambient conditions to keep density at the correct value [6] and some studies showed that BLYP water has a lower critical temperature and liquid densities than the corresponding experimental values [7–9]. Thus, the functional used also has a hand on the predicted value.

The distribution of the attribution is flat as shown for 3rd generation NNIP in Figure 1e. Hence, attribution has no impact on the density prediction. The NNIP at this generation, however, could already reproduce the structural properties of water.

We plotted the attribution vs bond length and added it to the Supplementary document. We found that even though the attributions on each atom are very low, the attribution on bonds shows interesting observations. The attributions on H-H and O-O bonds are low but those on O-H bond specifically are high. This is worth studying in detail later as attribution on atoms was sufficient to improve the accuracy of our potential through generations of active learning as explained in detail in the paper.

Action taken - We added a section on attribution on bond length in the Supplementary document.

7. Section 2.4: In figure 3a the NNIP is shown to under-predict the relative energy of nearly all 236 zeolites in relation to PBE-D3. In line 188, there is a claim that NNIP performed better than DFT and was demonstrated for 15 specific zeolites in Figure 3b. This characteristic was later attributed to the NNIP being trained on a higher level of hybrid functional. To back up this claim, the authors could calculate the remainder of the 236 zeolites using a different (i.e., better, or more appropriate) hybrid functional and reproduce Figure 3a to convince the reader.

Authors - Dispersion-corrected PBE is known to perform accurately for siliceous zeolites but SCAN+D3 functional shows better performance than PBE + D3 on few experimental zeolites with a mean absolute error (MAE) of around 1kJ/mol less [10]. Further, an NNIP trained on SCAN data has a lower MAE for the experimental zeolites than the PBE + D3

functional [10]. Similar to our model, the discrepancy with DFT-PBE-D3 results is a consequence of our and ref 23’s NNIPs reproducing accurately their higher level DFT training data.

8. Figure 3: Formation energies were compared relative to that of alpha-quartz. I assume the NNIP results were normalized by NNIP alpha-quartz and the PBE-D3 results were normalized by PBE-D3 alpha-quartz. If true, how does the reader know that the NNIP gets the energetics, structure, forces, angles, bond lengths, etc. of quartz correct? By comparing relative energies, are the data being obscured or misrepresented?

Authors - Our NNIP is used to obtain equilibrium structure properties of alpha-quartz to show that our NNIP gets its structure, angles, and bond lengths correct.

There are no absolute energies that we can compare since the zero energy levels among DFT functionals and also ML potentials are arbitrary. Rather, the usual practice is to compare energies concerning some chosen reference compound, in our case alpha-quartz. Further, we showed our model can reproduce the structure of alpha-quartz quite well, thus the data is not misrepresented based on prediction errors of alpha-quartz.

Actions taken - We obtained different bond lengths and bond angles in alpha-quartz at room temperature and compared them to experimental and DFT data. This will be added to the supplementary document.

9. Line 250-255: The sentence is difficult to understand and interpret at best. I’m confused as to how this makes the point of which mechanism occurs here. Maybe clarity could come from labels of attacking, leaving, etc. in Figure 5 and significant rewording.

Authors – DFT studies with implicit solvent effect have proposed two different types of mechanisms behind silicate dimerization at neutral conditions. One involves an S_N2 backside reaction [11], where the hydrogen from the attacking oxygen is transferred to the leaving oxygen at the back-side of the silicon attacked, through a chain exchange as shown in Figure 5a. Another mechanism, which has broader acceptance involves a lateral flank-side substitution reaction [12]. Here, the leaving oxygen lies on the lateral side of the silicon atom attacked and takes the hydrogen from the attacking oxygen concurrently to the O-Si bond formation

and leaves forming a water molecule as shown in Figure 5c. To test both mechanisms, we did umbrella sampling for both of them varying the choice of the leaving oxygen. For the S_N2 backside mechanism, we found that instead of going through a chain of exchange, the attacking oxygen loses its hydrogen to a surrounding water molecule to form a H_3O^+ , and the leaving oxygen leaves as OH^- forming a product state with energy higher than the reaction state. The H_3O^+ and the OH^- need to recombine to form our required product. However, due to the high energy difference and higher activation energy of 170 kJ/mol, the product tends to go back to the reaction state more instead of moving forward. This mechanism is shown in Figure 5b. However, when having the lateral oxygen as the leaving group, we saw the reaction playing out exactly as described in the lateral flank-side mechanism along with a neutral product which has almost similar energy to the reaction phase and a lower activation energy of 103 kJ/mol. Thus, we concluded that the dimerization in an aqueous solution at neutral conditions occurs through a lateral flank-side attack mechanism with pentavalent silicon as a transition state.

Actions taken – We have added the labels of attacking OH and leaving OH in Figure 5. We have added the free energy profile of the S_N2 mechanism in the Supplementary document. We also changed the wording of section 2.5.3 Silicate dimerization.

We next simulated a dimerization reaction of two silicate monomers in a neutral solution with 100 water molecules at 300K. We ran umbrella sampling on a reaction coordinate, $d_1 - d_2$ varying it from -5 Å to 5 Å with an increment of 0.05 Å. d_2 represents the bond distance between attacking oxygen and the attacked silicon, whereas d_1 represents the distance between the leaving oxygen and the attacked silicon. The attacking oxygen, attacked silicon, and the leaving oxygen atoms for the S_N2 and the lateral attack mechanisms are shown in Figure 5. The S_N2 mechanism we observed differed from the predicted one in explicit solvent. We observed the formation of H_3O^+ and OH^- which results in products with energy 79 kJ/mol higher than the reactants and the activation energy is also higher (180 kJ/mol) than the lateral attack mechanism. The free energy profile of the S_N2 mechanism is shown in Supplementary Figure 2. Hence, instead of the ions coming together to form the desired product, they tend to go back to the reaction state as shown in Figure 5a-b. However, in the case of the lateral attack mechanism, the reaction in condensed solvent follows

the proposed mechanism in continuum solvent [13–15], where the leaving oxygen leaves with the hydrogen from the attacking oxygen (Figure 5c). This helps us to conclude that silicate dimerization in an aqueous solution at neutral conditions occurs through a lateral flank-side attack mechanism with pentavalent silicon as a transition state. Now from the free energy profile of the lateral mechanism in Figure 4e, we see that at the coordinate close to 5 Å, the product dimer is formed. The reactant state with two monomers is at the coordinates between -3 and -5 Å where the reaction energy profile is almost flat. The product has 0.2 kJ/mole more energy than the reactant phase. In a DFT study with an implicit solvent effect, the authors found the product has an energy 9 kJ/mole higher than the reactant state with an activation energy of 129 kJ/mole [13]. We obtained the activation energy as 103 kJ/mole which can be trusted more as we used water solvent explicitly. From the plot of the coordination between the attacking oxygen with the transferring hydrogen and that of the leaving oxygen with the transferring hydrogen in Figure 4e, we further observed that at the transition state, the hydrogen leaves the attacking oxygen with a drop in their coordination value and moves to the leaving oxygen with an increase in their coordination value almost immediately as expected in the predicted lateral flank-side mechanism.

10. Atomic charge can play a key role in the dynamics of water and water/silica interactions. The NNIP does not include charges to the best of my understanding. Notably, the authors chose to address the acidic (low pH) chemical pathway for dimerization occurring through neutral reactant species. In the context of NNIP transferability, a discussion of the role of charge and the implications of NNIP not including charge is warranted. If NNIP does include charge, I missed that fact, and a discussion of how charge is included is warranted.

Authors - Our current NNIP model learns the charge implicitly from the configurations. There is no explicit addition of charge interactions or training on atomic charges. The training configurations are not all neutral because we don’t need a background charge correction in molecular clusters. Also, the whole cluster could be neutral and still contain charge separation. Anions such as OH⁻ and silicate or cations like Na⁺ are present, and this implicitly exposed our model to interactions between atomic charges. Because there are no redox reactions at play in this system, each chemical environment can be tied directly with a certain

set of atomic charges and our model learns those implicitly. In a system where oxidation and reduction are possible, that is no longer the case and it is often advantageous to add explicit prediction of point charges, such as in CHGNet [16] or point-charged NNIP models like ANI [17].

11. Lines135: Incomplete sentence, wording?

Actions taken – we changed the line 135. Benchmarking force error is not sufficient to quantify the quality of ML potentials [2] and simulation-based statistics should be used to evaluate model performance in production. Hence, we begin by testing the properties of liquid water.

12. Missing citations: Line81: “RDkit”, Line 86: “Orca”, line 319 “Packmol”, line 319: “Ovito”, line 358: “Pymatgen”

Actions taken - We have added references.

13. Line 168, 172: “...NQE...” do you mean QNE as abbreviated earlier?

Authors - Yes, it is the same as the QNE abbreviated earlier. We have corrected the sentence in the manuscript.

Actions taken – We changed the line 172.

References

1. Schwalbe-Koda, D., Tan, A. R. & Gómez-Bombarelli, R. Differentiable sampling of molecular geometries with uncertainty-based adversarial attacks. *Nature Communications* **12**, 1–12. <https://www.nature.com/articles/s41467-021-25342-8> (Aug. 2021).
2. Fu, X. *et al.* Forces are not Enough: Benchmark and Critical Evaluation for Machine Learning Force Fields with Molecular Simulations. <https://arxiv.org/abs/2210.07237v1> (Oct. 2022).
3. Liu, J., Lan, J. & He, X. Toward High-level Machine Learning Potential for Water Based on Quantum Fragmentation and Neural Networks. *Journal of Physical Chemistry A*

- 126**, 3926–3936. <https://pubs.acs.org/doi/full/10.1021/acs.jpca.2c00601> (June 2022).
4. Yao, Y. & Kanai, Y. Nuclear Quantum Effect and Its Temperature Dependence in Liquid Water from Random Phase Approximation via Artificial Neural Network. *Journal of Physical Chemistry Letters* **12**, 6354–6362. <https://pubs.acs.org/doi/full/10.1021/acs.jpcllett.1c01566> (July 2021).
 5. Gaiduk, A. P., Gustafson, J., Ois Gygi, F. & Galli, G. First-Principles Simulations of Liquid Water Using a Dielectric-Dependent Hybrid Functional. *J. Phys. Chem. Lett* **9**, 3068–3073. <https://pubs.acs.org/sharingguidelines> (2018).
 6. Yoo Willow, S., Zeng, X. C., Xantheas, S. S., Kim, K. S. & Hirata, S. Why Is MP2-Water "Cooler" and "Denser" than DFT-Water? <https://pubs.acs.org/sharingguidelines> (2016).
 7. Todorova, T., Seitsonen, A. P., Irg Hutter, J., Kuo, I.-F. W. & Mundy, C. J. Molecular Dynamics Simulation of Liquid Water: Hybrid Density Functionals †. <https://pubs.acs.org/sharingguidelines> (2006).
 8. McGrath, M. J. *et al.* Simulating Fluid-Phase Equilibria of Water from First Principles †. <http://www.llnl.gov/> (2006).
 9. McGrath, M. J. *et al.* Isobaric-isothermal monte carlo simulations from first principles: Application to liquid water at ambient conditions. *ChemPhysChem* **6**, 1894–1901 (Sept. 2005).
 10. Erlebach, A. *et al.* A reactive neural network framework for water-loaded acidic zeolites. <https://arxiv.org/abs/2307.00911v3> (July 2023).
 11. Pereira, J. C., Catlow, C. R. & Price, G. D. Silica condensation reaction: an ab initio study. *Chemical Communications*, 1387–1388. <https://pubs.rsc.org/en/content/articlelanding/1998/cc/a801816b> (Jan. 1998).
 12. Elanany, M. *et al.* A quantum molecular dynamics simulation study of the initial hydrolysis step in sol-gel process. *Journal of Physical Chemistry B* **107**, 1518–1524. <https://pubs.acs.org/doi/full/10.1021/jp026816z> (Feb. 2003).

13. Trinh, T. T., Jansen, A. P. & Van Santen, R. A. Mechanism of oligomerization reactions of silica. *Journal of Physical Chemistry B* **110**, 23099–23106. <https://pubs.acs.org/doi/full/10.1021/jp0636701> (Nov. 2006).
14. Zhang, X. Q., Trinh, T. T., Van Santen, R. A. & Jansen, A. P. Mechanism of the initial stage of silicate oligomerization. *Journal of the American Chemical Society* **133**, 6613–6625. <https://pubs.acs.org/doi/full/10.1021/ja110357k> (May 2011).
15. Schaffer, C. L. & Thomson, K. T. Density functional theory investigation into structure and reactivity of prenucleation silica species. *Journal of Physical Chemistry C* **112**, 12653–12662. <https://pubs.acs.org/doi/full/10.1021/jp066534p> (Aug. 2008).
16. Deng, B. *et al.* CHGNet as a pretrained universal neural network potential for charge-informed atomistic modelling. *Nature Machine Intelligence* / **5**, 1031–1041. <https://doi.org/10.1038/s42256-023-00716-3> (2023).
17. Gao, X., Ramezanghorbani, F., Isayev, O., Smith, J. S. & Roitberg, A. E. TorchANI: A Free and Open Source PyTorch-Based Deep Learning Implementation of the ANI Neural Network Potentials. *Journal of Chemical Information and Modeling* **60**, 3408–3415. <https://doi.org/10.1021/acs.jcim.0c00451> (July 2020).

REVIEWER COMMENTS

Reviewer #1 (Remarks to the Author):

My main concern regarding the initial manuscript revolved around the novelty of the methodological contribution, which has now been clarified by the authors: the machine learning force field (MLFF) "PaiNN" is used without modification, but a novel active learning scheme allows the selection of training points. In light of this, I wonder about the decision to rebrand the PaiNN potential to "NN-based interatomic potentials (NNIP)".

The improved datasets supposedly enable transferability from molecular clusters (training data) to periodic systems. This allows the MLFF to model the silicate dimerization reaction with explicit water solvent. As another reviewer has pointed out about the initial manuscript, the role and effectiveness of the active learning scheme for this application remains unclear from the revised manuscript. Maybe an ablation study that demonstrates the performance of the PaiNN architecture with and without the new sampling scheme should be included to discern the impact of the new sampling method?

Overall, the revised manuscript does not sufficiently articulate and define the novelty of the contribution of this work in comparison to existing research. The significance of the methodological contribution remains unclear in my opinion.

Reviewer #2 (Remarks to the Author):

The authors have addressed satisfactorily reviewers comments. The manuscript could be published as is.

Reviewer #3 (Remarks to the Author):

The authors have sufficiently addressed my earlier questions and comments on the manuscript. I thank the authors for their thorough response. I recommend the manuscript for publication in its current state.

Reply to Reviewers

Reviewer 1

Comments

My main concern regarding the initial manuscript revolved around the novelty of the methodological contribution, which has now been clarified by the authors: the machine learning force field (MLFF) “PaiNN” is used without modification, but a novel active learning scheme allows the selection of training points. In light of this, I wonder about the decision to rebrand the PaiNN potential to “NN-based interatomic potentials (NNIP)”.

Authors - The reviewer is absolutely correct that the novelty of this work is not in the neural network architecture, of which there are many and PAINN is just one instance. Our main methodological innovation is in the active learning strategy, namely in the use of attribution techniques to isolate small regions of high NNIP uncertainty in the simulation for active learning. The choice of NN potential architecture is irrelevant to that innovation, and our approach is compatible with other state-of-the-art architectures like MACE, M3GNet or CHGNet[1–3]. In addition, we have not rebranded anything. We refer to the NNIP we used sometimes by the architecture name (PAINN) and sometimes by the more general term “NN-based interatomic potentials (NNIP)” to highlight that the methodological innovation is irrespective of the choice of NNIP.

The improved datasets supposedly enable transferability from molecular clusters (training data) to periodic systems. This allows the MLFF to model the silicate dimerization reaction with explicit water solvent. As another reviewer has pointed out about the initial manuscript, the role and effectiveness of the active

learning scheme for this application remain unclear from the revised manuscript. Maybe an ablation study that demonstrates the performance of the PaiNN architecture with and without the new sampling scheme should be included to discern the impact of the new sampling method?

Authors - The dataset provably enables transferability from clusters to periodic systems. Unlike other NNIPs, which in addition to an architecture have also released a trained model (MACE, M3GNet, CHGNet) there is no pre-trained model for PAINN, it is just an architecture for which one fits the data they have[4]. A pre-trained potential based on MACE architecture came out last year, while this paper was under review, that can somewhat capture the interaction between silica and water interface, but not their chemical reactivity[5].

To highlight how the ML potential improves with the added data, the manuscript now compares the base generation NNIP trained on our preliminary dataset of 130K molecular geometries to our final generation NNIP, trained on the additional data obtained through our uncertainty attribution-based active learning. The base generation NNIP shows an accuracy of 1.15 kcal/mol/Å on the test dataset. However, benchmarking force accuracy is not good enough[6].

We obtained the O-O radial distribution function (rdf) from a molecular dynamics simulation at 300K with our base generation NNIP. Supplementary Figure 3 shows that the base generation NNIP could not produce accurate O-O rdf. However, after attribution-based active learning, our final generation NNIP can predict the structural properties of water accurately. Active learning has proven to improve the efficiency of Neural network-based potentials in many cases[7–10]. We have further shown in Figure 1c-e how our attribution-based active learning helps us choose molecular clusters which the NNIP is most uncertain about and how generation-wise, it helps reducing the uncertainty for our NNIP thus improving its efficiency. We further added supplementary figure 4 which shows that DFT calculation time varies exponentially with the number of atoms in a molecule cluster. Our active learning method helps to extract molecular clusters of size 10-300 atoms from periodic systems of size 700-1200 atoms, thus also saving a huge amount of DFT time.

Actions taken - We have added supplementary figures 3 and 4.

Overall, the revised manuscript does not sufficiently articulate and define the

novelty of the contribution of this work in comparison to existing research. The significant of the methodological contribution remains unclear in my opinion.

Authors - Our methodological contribution does not involve creating a new force field architecture, as the reviewer correctly states. It is unfortunate if the previous version of the manuscript somehow suggested that. Our methodological innovation is in the use of uncertainty attribution through auto-differentiation. We thus introduce a novel method of active learning that uses local atomic environments by generating a per-atom force uncertainty metric. This active learning method helps us extract small molecular fragments with high uncertainty from large simulation periodic boxes. Running quantum chemical calculation on smaller fragments is more affordable, and even allows us to use molecular DFT with a hybrid functional for higher accuracy in chemical reactivity. This is particularly useful because it applies to any NNIP architecture, not just the one we arbitrarily picked in this manuscript (PAINN). Training reactive potentials for solids using hybrid DFT is also novel. Thus, our methodological contribution is a new local atomic environment-based active learning method. Further, we utilize that methodological novelty to answer a scientific question accurately and rigorously. Since the NNIP can simulate chemical bond breaking and formation accurately in silica-water configurational space, we simulate silicate dimerization reaction with explicit water solvent in a neutral medium and gain novel insight into the mechanism.

References

1. Batatia, I., Kovács, D. P., Simm, G. N. C., Ortner, C. & Csányi, G. MACE: Higher Order Equivariant Message Passing Neural Networks for Fast and Accurate Force Fields. *Advances in Neural Information Processing Systems* **35**, 11423–11436 (Dec. 2022).
2. Deng, B. *et al.* CHGNet as a pretrained universal neural network potential for charge-informed atomistic modelling. *Nature Machine Intelligence* / **5**, 1031–1041. <https://doi.org/10.1038/s42256-023-00716-3> (2023).
3. Chen, C. & Ong, S. P. A universal graph deep learning interatomic potential for the periodic table. *Nature Computational Science* 2022 2:11 **2**, 718–728. <https://www.nature.com/articles/s43588-022-00349-3> (Nov. 2022).

4. Schütt, K. T., Unke, O. T. & Gastegger, M. Equivariant message passing for the prediction of tensorial properties and molecular spectra. *arXiv preprint arXiv:2102.03150*. <http://arxiv.org/abs/2102.03150> (2021).
5. Batatia, I. *et al.* A foundation model for atomistic materials chemistry (2024).
6. Fu, X. *et al.* Forces are not Enough: Benchmark and Critical Evaluation for Machine Learning Force Fields with Molecular Simulations. <https://arxiv.org/abs/2210.07237v1> (Oct. 2022).
7. Schwalbe-Koda, D., Tan, A. R. & Gómez-Bombarelli, R. Differentiable sampling of molecular geometries with uncertainty-based adversarial attacks. *Nature Communications* **12**, 1–12. <https://www.nature.com/articles/s41467-021-25342-8> (Aug. 2021).
8. Kulichenko, M. *et al.* Uncertainty-driven dynamics for active learning of interatomic potentials. *Nature Computational Science* *2023 3:3* **3**, 230–239. <https://www.nature.com/articles/s43588-023-00406-5> (Mar. 2023).
9. Erlebach, A., Nachtigall, P. & Grajciar, L. Accurate large-scale simulations of siliceous zeolites by neural network potentials. *npj Computational Materials* *2022 8:1* **8**, 1–12. <https://www.nature.com/articles/s41524-022-00865-w> (Aug. 2022).
10. Erhard, L. C., Rohrer, J., Albe, K. & Deringer, V. L. Modelling atomic and nanoscale structure in the silicon–oxygen system through active machine learning. *Nature Communications* *2024 15:1* **15**, 1–12. <https://www.nature.com/articles/s41467-024-45840-9> (Mar. 2024).

REVIEWERS' COMMENTS

Reviewer #1 (Remarks to the Author):

In the revised manuscript, all raised issues were addressed in a satisfactory fashion.

Overall, the revised manuscript highlights the methodological advances of this work more clearly. It is well written and educationally valuable.